# Phosphodiesterases 4B and 4D Differentially Regulate cAMP Signaling in Calcium Handling Microdomains of Mouse Hearts

**DOI:** 10.3390/cells13060476

**Published:** 2024-03-08

**Authors:** Axel E. Kraft, Nadja I. Bork, Hariharan Subramanian, Nikoleta Pavlaki, Antonio V. Failla, Bernd Zobiak, Marco Conti, Viacheslav O. Nikolaev

**Affiliations:** 1Institute of Experimental Cardiovascular Research, University Medical Center Hamburg-Eppendorf, 20246 Hamburg, Germany; 2German Center for Cardiovascular Research (DZHK), Partner Site Hamburg/Kiel/Lübeck, 20246 Hamburg, Germany; 3UKE Microscopy Imaging Facility (UMIF), University Medical Center Hamburg-Eppendorf, 20246 Hamburg, Germany; 4Eli and Edythe Broad Center of Regeneration Medicine and Stem Cell Research, Department of Obstetrics, Gynecology, and Reproductive Sciences, Center for Reproductive Sciences, University of California, San Francisco, CA 94143, USA; marco.conti@ucsf.edu

**Keywords:** cAMP, FRET, microdomain, phosphodiesterase, calcium cycling

## Abstract

The ubiquitous second messenger 3′,5′-cyclic adenosine monophosphate (cAMP) regulates cardiac excitation-contraction coupling (ECC) by signaling in discrete subcellular microdomains. Phosphodiesterase subfamilies 4B and 4D are critically involved in the regulation of cAMP signaling in mammalian cardiomyocytes. Alterations of PDE4 activity in human hearts has been shown to result in arrhythmias and heart failure. Here, we sought to systematically investigate specific roles of PDE4B and PDE4D in the regulation of cAMP dynamics in three distinct subcellular microdomains, one of them located at the caveolin-rich plasma membrane which harbors the L-type calcium channels (LTCCs), as well as at two sarco/endoplasmic reticulum (SR) microdomains centered around SR Ca^2+^-ATPase (SERCA2a) and cardiac ryanodine receptor type 2 (RyR2). Transgenic mice expressing Förster Resonance Energy Transfer (FRET)-based cAMP-specific biosensors targeted to caveolin-rich plasma membrane, SERCA2a and RyR2 microdomains were crossed to PDE4B-KO and PDE4D-KO mice. Direct analysis of the specific effects of both PDE4 subfamilies on local cAMP dynamics was performed using FRET imaging. Our data demonstrate that all three microdomains are differentially regulated by these PDE4 subfamilies. Whereas both are involved in cAMP regulation at the caveolin-rich plasma membrane, there are clearly two distinct cAMP microdomains at the SR formed around RyR2 and SERCA2a, which are preferentially controlled by PDE4B and PDE4D, respectively. This correlates with local cAMP-dependent protein kinase (PKA) substrate phosphorylation and arrhythmia susceptibility. Immunoprecipitation assays confirmed that PDE4B is associated with RyR2 along with PDE4D. Stimulated Emission Depletion (STED) microscopy of immunostained cardiomyocytes suggested possible co-localization of PDE4B with both sarcolemmal and RyR2 microdomains. In conclusion, our functional approach could show that both PDE4B and PDE4D can differentially regulate cardiac cAMP microdomains associated with calcium homeostasis. PDE4B controls cAMP dynamics in both caveolin-rich plasma membrane and RyR2 vicinity. Interestingly, PDE4B is the major regulator of the RyR2 microdomain, as opposed to SERCA2a vicinity, which is predominantly under PDE4D control, suggesting a more complex regulatory pattern than previously thought, with multiple PDEs acting at the same location.

## 1. Introduction

3′,5′-Cyclic adenosine monophosphate (cAMP) is a ubiquitous second messenger responsible for intracellular signal transduction and thus involved in the regulation of numerous independent biological processes [1,2,3]. Acting in discrete subcellular microdomains, cAMP is responsible for positive chronotropic, inotropic and lusitropic effects in the myocardium [4].

Compartmentation of cAMP signaling in mouse cardiomyocytes (CMs) is an important factor in maintaining the accuracy of receptor-mediated responses. It is achieved, for example, by the formation of signaling complexes that bring together cAMP effectors, such as cAMP-dependent protein kinase (PKA), with its target proteins and degrading enzymes phosphodiesterases (PDEs) using scaffolding proteins that belong to the A-kinase-anchoring protein family [5,6]. Microdomains formed around Ca^2+^ handling proteins that are involved in the cardiac excitation-contraction coupling (ECC), are usually regulated independently from the bulk cytosol [7,8]. These subcellular microdomains, which are also often referred to as nanodomains, consist of receptors, A-kinase-anchoring proteins, protein kinases, phosphatases and cyclic nucleotide PDEs [9,10]. The latter enzymes are critical for shaping local cAMP pools in various types of mammalian cells [11,12]. More than 100 isoforms belong to the PDE superfamily which can be classified into 11 families (PDE1–PDE11) based on structure and substrate specificity [11,13]. PDE4 is the predominant PDE family in rodent CMs and is encoded by 4 genes (PDE4A, PDE4B, PDE4C and PDE4D), whereas only PDE4A, PDE4B and PDE4D are expressed in CMs [14,15].

Increased levels of cAMP due to β-adrenergic (β-AR) stimulation lead to activation of PKA, which phosphorylates several downstream targets that are involved in the ECC, such as the L-type calcium channel (LTCC), its associated small regulatory G-protein called Rad, phospholamban (PLN) and calcium release channels called ryanodine receptors (RyR2). β-AR/cAMP/PKA dependent phosphorylation of Rad and PLN relieves the inhibition of LTCC and sarco/endoplasmic reticulum Ca^2+^-ATPase (SERCA2a), respectively, leading to positive inotropic and lusitropic effects, with positive inotropic effect being additionally augmented by RyR2 phosphorylation of RyR2at serine 2808 PKA consensus site [16,17,18]. Alterations in the regulation of the microdomains formed around calcium handling proteins result in severe heart diseases such as arrhythmias and heart failure [19,20,21].

Live cell imaging based on Förster Resonance Energy Transfer (FRET) is a powerful tool to visualize cyclic nucleotide dynamics in living cells and tissues in real time [7,8,9,13,22,23,24]. Targeted versions of FRET-based biosensors have recently enabled uncovering changes in cAMP concentrations with a high temporal and spatial resolution directly in various microdomains associated with calcium homeostasis to provide a better understanding of subcellular signaling events in CMs [25,26,27]. For example, we could show that heart failure leads to redistribution of multiple PDE isoforms between various plasma membrane and SR microdomains, thereby altering cardiac contractility, β-AR dependent regulation of calcium homeostasis and loss of PDE4-mediated protection from RyR2-dependent arrhythmias [25,26,27].

Here, we systematically investigated the specific roles of PDE4B and PDE4D in regulating local cAMP signaling in the vicinity of three distinct calcium handling proteins using a combination of three targeted CM-specific FRET sensor transgenic mice with PDE4 knockout models. We were able to confirm several previously reported findings and uncovered the new role of PDE4B which can directly control cAMP levels in the RyR2 microdomain and their pro-arrhythmic potential in addition to the known role of PDE4D. Therefore, the mechanisms of local cAMP regulation and their functional implications appear to be more complex than previously thought.

## 2. Materials and Methods

### 2.1. Animals

All animal procedures were approved by local animal welfare authority BJV Hamburg (approval number ORG1010) and conformed to the guidelines from Directive 2010/63/EU of the European Parliament on the protection of animals use for scientific purposes. We used both female and male mice, randomization and blinding regarding their sex and genotypes. The following mouse lines were used for this project: pmEpac1 (encoding the cAMP biosensor targeted to caveolin-rich plasma membrane domains) [25], Epac1-PLN (which is a cAMP biosensor for the vicinity of SERCA2a) [26], Epac1-JNC (cAMP biosensor for RyR2 vicinity) [27], PDE4B knockout (KO) [28] and PDE4D-KO mice [29]. FRET sensor mice were bred with PDE4-KO lines and kept on mixed FVB/N1; C57/BL6J background. All results obtained for knockout animals were directly compared to wildtype littermates and no data exclusion was performed. For CM isolation and whole heart perfusion experiments, animals were sacrificed at the age of 8–20 weeks.

### 2.2. Cardiomyocyte Isolation

Mice were euthanized by cervical dislocation in deep isoflurane anesthesia. Hearts were rapidly extracted and adult ventricular CMs were isolated via retrograde Langendorff perfusion, as previously described [30].

### 2.3. Live Cell Imaging

FRET measurements and data analysis were performed with isolated CMs plated onto laminin coated glass coverslides, using an inverted fluorescent microscope, according to a previously published protocol [31]. Measurements were performed 2–10 h after cell isolation. Micro Manager software (version 1.4.5, Vale Lab, San Francisco, CA, USA) was used for image acquisition. For chemical myocyte detubulation, cells were incubated with 1.5 mol/L formamide solution for 15 min at room temperature, resulting in an osmotic shock-induced detubulation [23,32].

### 2.4. Single Cell Contractility Measurements

After CM isolation, cells were washed with contractility measurement buffer containing NaCl, 149 mmol/L; KCl, 1 mmol/L; MgCl_2_
*×* 6H_2_O, 1 mmol/L; HEPES, 5 mmol/L; glucose, CaCl_2_, 1 mmol/L, and incubated for at least 90 min at 37 °C and 5% CO_2_. CM suspension was added into a measuring chamber and pharmacologically stimulated with 100 nmol/L Isoproterenol (ISO). Electrical stimulation was performed for 4 ms at 0.5 s^−1^ and 15.0 V. Contractile responses were evaluated by the optical sarcomere length measurement method (IonOptix, Westwood, MA, USA).

### 2.5. Immunofluorescence Staining and STED Microscopy

Adult mouse CMs, plated on round glass coverslides, were fixed with ice-cold ethanol for 20 min at −20 °C. After three washing steps with phosphate buffer saline (PBS), CMs were blocked by incubation with blocking buffer (consisting of PBS with 10% fetal calf serum and 0.3% of Triton X-100) for 2 h at room temperature in the dark. Cells were incubated with primary antibodies against PDE4B (kindly provided by George Baillie, dilution 1:200) and RyR2 (Sigma Aldrich, Taufkirchen, Germany, dilution 1:300) over night at 4 °C. After washing thrice, secondary antibody incubation was performed in the dark using Alexa Fluor 594 Donkey Anti-Sheep (Thermo Fisher Scientific, Bremen, Germany, dilution: 1:250) and Goat Anti-Rabbit IgG-Abberior Star RED (Sigma Aldrich, dilution 1:250), respectively, for 2 h at room temperature. STED and corresponding confocal microscopy were carried out in sequential line scanning mode using an Abberior STED microscope (Göttingen, Germany). The setup was built on a Nikon Ti-E microscope body equipped with autofocus and a 60× (NA 1.4) P-Apo oil immersion objective. Two pulsed lasers were used for excitation at 561 and 640 nm and near-infrared pulsed laser (775 nm) for depletion. The detected fluorescence signal was directed through a variable sized pinhole (set to match 1 Airy at 640 nm) and detected by novel state of the art avalanche photo diodes (APDs) with appropriate filter settings for Cy3 (595–635 nm) and Cy5 (615–755 nm). Images were recorded with a dwell time of 5 µs and the pixel size was set to be 20 × 20 nm. The images were taken in 2D-STED mode. The acquisitions were carried out in time gating mode i.e., with a time gating width of 8 ns and a delay of 781 ps for both the red and far red channel.

### 2.6. Langendorff-Perfused Whole Heart Stimulation

Animals were sacrificed at the age of 8–20 weeks in isoflurane anesthesia and hearts were rapidly extracted into ice-cold PBS. After cannulation, hearts were perfused at 37 °C with Langendorff perfusion buffer (NaCl, 118 mmol/L; KCl, 4.7 mmol/L; MgSO_4_
*×* 7H_2_O, 0.8 mmol/L; NaHCO_3_, 25 mmol/L; KH_2_PO_4_, 1.2 mmol/L; glucose, 5.0 mmol/L; sodium pyruvate, 110 mmol/L; CaCl_2_, 2.5 mmol/L). The buffer was carbogenated (5% CO_2_ in O_2_) throughout the whole experiment to prevent calcium from precipitating. Langendorff measurements were performed as described previously [27]. Hearts were equilibrated for 20 min with programmed pacing at 600 beats per minute (cycle length 100 ms). To detect ventricular arrhythmias, ventricular burst pacing protocol (5 s at S1S1: 50–10 ms with 10 ms stepwise reduction) was used. Ventricular tachycardias (VTs) were defined as ≥4 consecutive premature ventricular complexes. Burst protocol was performed three times before and three times after perfusion with 100 nmol/L isoproterenol (ISO).

### 2.7. Immunoblot Analysis and Co-Immunoprecipication

Shock-frozen mouse hearts or CM pellets were homogenized in lysis buffer (containing Tris, 30 mmol/L; EDTA, 1 mmol/L; NaCl, 150 mmol/L; NP-40, 1 Vol%; 20% SDS solution, 0.1 Vol%, Roche cOmplete protease and phosphatase inhibitor cocktail tablets). Protein concentrations were quantified with a bicinchoninic acid assay. Approximately 20–45 µg of protein was diluted in lysis buffer including 3x SDS Stop buffer (containing 200 mmol/L Tris; 20% SDS solution, 6 Vol%, glycerol, 15 Vol%; β-Mercaptoethanol, 10 Vol%). Samples for detection of PDE2A (FabGennix, Frisco, TX, USA, dilution 1:500), PDE3A (kindly provided by Chen Yan, dilution 1:1000), PDE4A (abcam, Cambridge, UK, dilution 1:500), PDE4B (abcam, dilution 1:1000) and PDE4D (abcam, dilution 1:1000) were boiled for 5 min at 95 °C. Samples for pPLN (pSer16, Badrilla, Leeds, UK, dilution 1:5000; total PLN, Badrilla, dilution 1:2500) and pRyR2 (pSer2808, Badrilla, dilution 1:2500; total RyR2, Sigma Aldrich, dilution 1:5000) detection were not boiled. Proteins were separated according to their size by sodium dodecyl sulfate polyacrylamide gel electrophoresis. Separated proteins were transferred onto a nitrocellulose membrane and were analyzed using specific antibodies. For co-immunoprecipitation, adult mouse hearts expressing the RyR2 targeted Epac1-JNC biosensor or wildtype mouse hearts were lysed in 200 μL co-IP lysis buffer (Tris, 10 mmol/L; NaCl, 120 mmol/L; EDTA, 0.5 mmol/L; Triton X-100, 0.5%; pH 7.4; protease and phosphatase inhibitors), and either the RyR2 complex was pulled down using GFP-Trap^®^ beads or RyR2 was directly immunoprecipitated using total RyR2 antibody (Sigma Aldrich) and protein A/G beads (Santa Cruz Biotechnology, Dallas, TX, USA) with subsequent RyR2, PDE4B and PDE4D immunoblot detection as previously described [27].

### 2.8. Statistics

GraphPad Prism 9 (version 9.3.0, GraphPad Software, Boston, MA, USA) was used for statistical analysis. Data are presented as mean ± SEM. Normal distribution was tested by the Kolmogorov–Smirnov test, and normally distributed data were analyzed by one-way ANOVA with or without Sidak multiple comparison test, whereas Kruskal–Wallis test was used for skewedly distributed data. When multiple cells isolated from the different animal were compared, mixed ANOVA followed by Wald χ^2^-test was used. Differences were considered significant at *p* < 0.05. No data exclusion has been made. Only cells which did not respond to initial β-adrenergic (ISO) stimulation were not further measured.

## 3. Results

### 3.1. Impact of PDE4B and PDE4D Ablation on PDE Expression

To analyze whether global deficiency of PDE4B and PDE4D in adult mice causes CM specific changes in the expression of other major PDE subfamilies such as PDE2A, PDE3A, PDE4A, PDE4B vs. PDE4D, adult hearts from PDE4B-WT, PDE4B-KO, PDE4D-WT and PDE4D-KO mice were harvested and analyzed using immunoblots. No compensatory changes in PDE2A, PDE3A or PDE4A expression were observed in neither PDE4B-KO nor in PDE4D-KO mouse hearts (Appendix A), suggesting that genetic deletion of these PDE families does not lead to altered expression of other major PDEs. To assess if individual knockouts can cause possible compensation of cAMP hydrolyzing by the remaining PDE4 subfamily, we measured catalytic activity of PDE4B and PDE4D immunoprecipitated from wildtype, PDE4B-KO and PDE4D-KO hearts. We could detect no significant change of PDE4B activity in PDE4D-KO mice, whereas in PDE4B-KO hearts, there was a slight but significant increase in PDE4D activity (Appendix A).

### 3.2. PDE4B and PDE4D Are Both Involved in the Regulation of cAMP Dynamics in the Caveolin-Rich Plasma Membrane

To investigate the impact of PDE4B and PDE4D on cAMP dynamics at the caveolin-rich plasma membrane microdomains, real-time FRET measurements were performed using CMs freshly isolated from transgenic mice expressing the highly sensitive membrane targeted FRET-based biosensor pmEpac1 in wildtype vs. PDE4B or PDE4D null background. Although not directly co-localized with the LTCCs, the pmEpac1 biosensor is a suitable tool to monitor cAMP dynamics in the caveolin-rich plasma membrane microdomains including T-tubules where these channels are located at high density [33].

Local cAMP accumulation in response to β-AR stimulation by ISO was significantly enhanced in both PDE4B and PDE4D knockout CMs as compared to the maximal response evoked by the combination of the broadband PDE inhibitor 3-isobutyl-1-methylxanthine (IBMX) combined with the adenylyl cyclase activator forskolin (Figure 1A,B). Here and in the following experiments, a decrease of FRET ratio represents an increase of intracellular cAMP.

From these data, there is a clear indication that both PDE subfamilies are involved in the regulation of the plasma membrane microdomain, most likely by regulating local cAMP hydrolysis. To confirm that this effect is not due to increased cAMP synthesis of the knockout cardiomyocytes, ISO-prestimulated CMs were acutely treated with a high concentration of the β-AR antagonist propranolol. Prestimulation with ISO leads to an accumulation of cAMP within the microdomain, while propranolol treatment causes an instantaneous inhibition of cAMP production, after which microdomain-specific relaxation kinetics can be measured. The half-maximum cAMP degradation time τ_1/2_ (determined by a monoexponential fit of the respective FRET traces) after adding propranolol reflects the rate of local cAMP degradation which should directly correlate with microdomain-specific PDE activity [34]. Interestingly, τ_1/2_ was significantly enhanced in both knockouts (Figure 1C,D), suggesting that PDE4B and PDE4D in concert regulate local cAMP dynamics.

### 3.3. SERCA2a Microdomain Is Predomimantly Regulated by PDE4D

To study the effects of PDE4B and PDE4D subfamilies on cAMP dynamics in the SERCA2a microdomain, we performed FRET measurements using CMs from transgenic mice expressing the Epac1-PLN biosensor crossed with PDE4B- and PDE4D-deficient mice. The ratio between the ISO induced cAMP accumulation compared to the maximum cAMP concentration showed significant changes in PDE4D-deficient CMs, whereas no changes could be detected in PDE4B-knockout cells (Figure 2A,B). Increased cAMP levels in the PDE4D-KO CMs were most likely based on a locally decreased PDE activity, since τ_1/2_ during propranolol treatment was increased by ~40%, as compared to the respective wildtype littermates (Figure 2C,D). Still, detectable decreases of cAMP in the absence of PDE4D suggest that cAMP either diffuses out of this microdomain and/or PDE4D is not the only PDE subfamily involved in the SERCA2a microdomain regulation.

Downstream effects of PDE4D ablation on the SERCA2a microdomain regulation were further analyzed in terms of PKA-dependent phosphorylation of its inhibitory protein regulator PLN at Ser16. In PDE4D-KO left ventricular tissue, PLN phosphorylation normalized on the total-PLN protein expression was significantly increased, whereas no changes could be detected in PDE4B hearts (Figure 2E). This was also the case in isolated CMs stimulated with ISO. Here, much stronger phosphorylation could be detected in PDE4D-KO cells than in PDE4B-KO cells as compared to their littermate controls (Figure 2F). To analyze possible redundancy of PDE4B and PDE4D, we used the previously established selective PDE4B inhibitor GSK16 [34] in PDE4D-KO CMs which had no additional effect in this setting (Appendix A).

### 3.4. cAMP Dynamics at the RyR2 Complex Are Predominantly Regulated by PDE4B

The Epac1-JNC FRET-based biosensor enables direct monitoring of cAMP levels in the RyR2 microdomain. Unexpectedly, no major increase in cAMP accumulation after stimulation with 100 nmol/L ISO could be detected in PDE4D-KO as compared to wildtype cells (Figure 3A,B).

Furthermore, τ_1/2_ after ISO and propranolol treatment was not significantly prolonged (Figure 3C,D). Strikingly, both treatment protocols indicated that PDE4B can affect cAMP levels at the RyR2 complex, since cAMP responses to ISO and τ_1/2_ calculated from propranolol signals were increased significantly by almost ~40 and ~120%, respectively (see Figure 3B,D). Next, we performed immunoblot analysis to gain deeper insights into the involvement of PDE4B and PDE4D in the regulation of the RyR2 microdomain. Left ventricular tissues from PDE4B-KO mice showed significantly increased Ser-2808 phosphorylation of the RyR2 as compared to PDE4B-WT tissues, whereas no significant differences could be detected for PDE4D deficient hearts, at least under basal conditions (Figure 3E). When analyzed in isolated CMs stimulated with ISO, RyR2 phosphorylation was significantly higher in PDE4B- but not in PDE4D-KO cells (Figure 3F).

Next, we looked at the cAMP responses to the general PDE4 inhibitor rolipram at the RyR2 microdomains of wildtype and PDE4B/4D-KO myocytes. Compatible with other data, rolipram effect after ISO stimulation was significantly reduced in PDE4B-KO cells, and this reduction was much more pronounced than that observed in PDE4D-KO cells (Figure 4A–C), confirming that cAMP dynamics in the vicinity of RyR2 is regulated by PDE4B >> PDE4D.

### 3.5. PDE4B Physically and Functionally Interacts with RyR2

To explore whether PDE4B can also physically interact with RyR2, we first performed co-immunoprecipitation experiments. As expected, we could detect both PDE4D and PDE4B in the RyR2 complex immunoprecipitated from mouse heart tissue via the Epac1-JNC biosensor and GFP-trap (Figure 4D). Likewise, PDE4B could be directly co-immunoprecipitated using total RyR2 antibody (Appendix A) as documented previously for PDE4D by us [27] and others [20]. Interestingly, when we applied the selective PDE4B inhibitor GSK16 after ISO to WT CMs, we could see much stronger response in Epac1-JNC biosensors expressing cells as compared to Epac1-PLN, suggesting tight functional association of this PDE4 isoform with RyR2 (Figure 4E,F and Appendix A).

To study functional effects of PDE4B and PDE4D, we analyzed arrhythmia susceptibility determined by quantifying extra beats in paced single CMs. Cells were isolated from PDE4B- and PDE4D-deficient mice as well as from the respective wildtype littermates and stimulated with 100 nmol/L ISO during pacing at 0.5 s^−1^ frequency. Extra beats were counted during a period of 60 s right after the ISO-induced maximum change of the sarcomere length. Almost every contraction was followed by an additional, spontaneous afterbeat in ISO stimulated PDE4B-KO cells (Figure 5 and Appendix A).

PDE4D-KO CMs also showed increased arrhythmia susceptibility but a significantly lower number of extra beats as compared to PDE4B-KO cells (Figure 5E) suggesting that both PDE4B > PDE4D are critical regulators of arrhythmia susceptibility, and that PDE4B activity might also be, at least in part, associated with the RyR2 microdomain. To analyze arrhythmia susceptibility in a more relevant context, we performed experiments in Langendorff perfused and electrically stimulated hearts of all genotypes with and without ISO stimulation. Here, we could detect a strong increase in ventricular tachycardia in PDE4B-KO hearts under ISO stimulation, as opposed to unchanged number of arrhythmias in PDE4D-KO hearts, at least under this stimulation protocol (Figure 6 and Appendix A).

To determine whether PDE4B is directly involved in the regulation of cAMP dynamics in the RyR2 microdomain, FRET experiments were performed to test the contribution of T-tubule associated PDE4B pools. An indirect regulation of cAMP dynamics in the RyR2 microdomain could be brought about by spatial proximity of T-tubular and SR membranes separated by the junctional gap of only ~15 nm, resulting in a detection of the well-established LTCC associated PDE4B activity [19] in the RyR2 complex. FRET experiments shown in Appendix A were performed in freshly isolated adult mouse CMs subjected to formamide-induced detubulation. The ratio between the ISO induced cAMP accumulation and the maximum cAMP concentration was still significantly increased in PDE4B-KO CMs, as was the τ_1/2_ value after propranolol treatment, whereas no significant changes in the cAMP levels could be detected in PDE4D deficient CMs (see Appendix A).

### 3.6. PDE4B Co-Localizes with RyR2

Finally, high resolution stimulated emission depletion (STED) microscopy was performed to determine the subcellular localization of PDE4B by specific immunofluorescence staining. Confocal and STED microscopy images of PDE4B and RyR2 stained CMs are shown in Figure 7A. The evaluation of the colocalization was limited due to the pixel size of the recorded STED microscopy images. In our case, one pixel represents 20 nm which approximately corresponds to the distance between the LTCC and the RyR2. This makes it impossible to completely separate the LTCC and RyR2 associated peaks of PDE4B signal. Therefore, colocalization was analyzed along the Z-lines and the distribution between directly overlapping and slightly shifted signals was quantified (see Figure 7B). In total, 84 Z-lines of 11 individual cells were analyzed resulting in 48.8% of the analyzed Z-lines showing a direct overlap between the fluorescence peak of PDE4B and RyR2, whereas 51.2% of the analyzed peaks were shifted by 1–2 pixels which corresponds to 20–40 nm (see Figure 7C,D). The distribution of almost 50:50 suggests that PDE4B might be localized in close proximity to both LTCC and RyR2.

## 4. Discussion

Ca^2+^ cycling is crucial for a proper cardiac excitation-contraction coupling, and several PDEs are involved in the regulation of Ca^2+^ homeostasis by controlling cAMP levels in the vicinities of LTCC, RyR2 and SERCA2a. The relative impact of PDE4B on the caveolin-rich plasma membrane has previously been reported in neonatal mouse CMs [34]. This study also detected somewhat increased RyR2 phosphorylation in PDE4B-KO mouse hearts but did not further analyze this phenomenon [34]. Another recent study has analyzed spatiotemporal regulation of PKA activity at RyR2 and SERCA2a using live cell imaging with targeted PKA activity reporters focusing on the role of inhibitory G-proteins and relative contributions of PDE2, PDE3 and PDE4 families [35]. However, the relative roles of individual PDE4B and PDE4D subfamilies in this process have not been systematically compared in adult CMs. Therefore, the present study focused on PDE4B and PDE4D subfamilies and used live cell imaging to measure and compare real time cAMP dynamics in the respective microdomains and to assess the contribution of individual PDE4 subfamilies by genetic ablation using a combination of knockout mice for both PDEs and three targeted cAMP biosensor expressing mouse lines. The conserved expression of PDE2A, PDE3A and other PDE4 subfamilies in mice with a global genetic deletion of PDE4B or PDE4D made it possible to analyze the impact of these two individual PDE4 subfamilies on local, microdomain-specific cAMP levels by targeted FRET based biosensors.

FRET microscopy with myocytes expressing pmEpac1, a biosensor for cAMP imaging in the caveolin-rich plasma membrane domains, showed functional involvement of both PDE4B and PDE4D in the regulation of cAMP signaling events which is compatible with a previous report showing that both PDEs co-immunoprecipitate with LTCC [19] (see Figure 1 and Figure 8). Comparable contribution of PDE4B to the caveolin-rich plasma membrane have previously been reported in neonatal mouse myocytes [34]. The fact that this finding could be now verified in adult mouse CMs is highly remarkable since T-tubules are absent in neonatal CMs [36]. Thus, reorganization of membrane structures and membrane protein relocations occurring upon T-tubule formation do not seem to affect the impact of PDE4B on the regulation of cAMP dynamics in the caveolin-rich membrane domains.

Using Epac1-PLN biosensor specifically targeted to the SERCA2a microdomain, we found that local cAMP levels are under the predominant control of the PDE4D subfamily (see Figure 2 and Figure 8), which also correlated with PLN phosphorylation at Ser-16. This finding is in line with a previous study which identified PDE4D (isoforms 3/8/9) as the major PDE4 subfamily physically and functionally associated with the SERCA2a/PLN signalosome [37]. This study also found significantly elevated levels of phosphorylated PLN in PDE4D-KO CMs, corroborating that our data are fully compatible with previously published results. Interestingly, they also found that basal RyR2 phosphorylation at Ser-2808 was very slightly but significantly reduced in PDE4D-KO [37]. In our experiments, even in the absence of PDE4D, cAMP levels kept decreasing after propranolol was added to ISO-prestimulated CMs. This could be due to the well-established PDE3A dependent regulation of the SERCA2a microdomain [38,39]. Due to limited capacity for breeding of multiple transgenic mouse lines, we were not able to analyze the individual contributions of PDE2A and PDE3A by genetic deletion in any of the three studied microdomains. Based on previous studies using family-selective PDE inhibitors, these PDEs contribute much less to overall cAMP level regulation that PDE4 enzymes, although they play important roles in the regulation of contractility and cross-talk between cyclic guanosine monophosphate and cAMP [9,25,26]. While being predominant on rodent myocytes, PDE4 is crucially involved in arrhythmias protection also in human myocytes, despite it contributing only 10–20% to the total cAMP hydrolysis [19,40]. This motivated us to closely look into the role of PDE4B and 4D in SR microdomains associated with increased arrhythmia susceptibility.

cAMP dynamics in the RyR2 complex was analyzed using a highly sensitive localized FRET biosensor Epac1-JNC which was generated by fusing Epac1-camps to junctin (JNC), a protein interacting directly with the RyR2 [27]. A previous study has so far identified only one PDE isoform present in the RyR2 complex belonging to the PDE4D family which is PDE4D3 [20]. Although PDE4B overexpression in transgenic mice led to ameliorated cardiac remodeling, no systematic investigation of its anti-arrhythmic effect was performed [41]. Our systematic live cell imaging analysis performed in PDE4B-KO CMs uncovered a decreased PDE activity within this complex, suggesting that this PDE can regulate local cAMP levels in addition to PDE4D (see Figure 3, Figure 4 and Figure 8). Even in the absence of either PDE4B or PDE4D, cAMP levels kept decreasing after propranolol application, suggesting possible involvement of other PDEs or equilibration of local cAMP levels between different microdomains. This could be due to some other PDEs present at a functionally relevant distance from RyR2, such as the remaining PDE4 subfamily or, for example, PDE8A, since it has been described that PDE8A deficient CMs show a “leaky” RyR2 phenotype [42]. Signaling events downstream of β-AR/cAMP and their contribution to the RyR2 microdomain regulation were further analyzed by looking at Ser-2808 phosphorylation of RyR2 which confirmed that PDE4B plays an important role in this process. PDE4B could be also co-immunprecipitated with RyR2 in addition to PDE4D (previously confirmed by us using the same protocol, see [27]). The PDE4B effect measured in the RyR2 microdomain could also theoretically originate from indirect effects of PDE4B located in close proximity, e.g., at T-tubular membrane around LTCC which is no more than ~15 nm away. However, it seems unlikely based on our experiments in CMs after formamide-induced osmotic detubulation, which showed a similar pattern in regard to cAMP signaling events (see Appendix A). This is further supported by super resolution STED microscopy, which compared to conventional microscopy shows up to 12-fold improved spatial resolution [43]. Together with FRET experiments in detubulated CMs and previous reports our STED data suggest that PDE4B might be present in both LTCC- and RyR2-associated microdomains.

We could verify the direct impact of PDE4B and PDE4D deficiency on arrhythmia susceptibility by performing analysis of extra beats measurements of single CMs as well as by measuring arrhythmia occurrence in intact Langendorff perfused hearts, either at basal level or upon β-AR stimulation (see Figure 5 and Figure 6). In our study, PDE4B-KO showed significantly more extra beats than PDE4D-KO in isolated cells. In single myocytes, all extra beats occurred at the earliest, at 500 ms after the cell was electrically stimulated, reminiscent of delayed afterdepolarizations. A previous study reported comparable increase in the number of extra beats in both PDE4B- and PDE4D- deficient mice [19]. However, the number of spontaneous beats we could detect in PDE4B-KO CMs was much higher than in the previous study, while the number of extra beats in PDE4D-KO CMs was rather comparable. This could be explained by a slightly different experimental setup. In contrast to previously published “pulse” stimulation with 100 nmol/L ISO for 15 s, we used a continuous ISO stimulation and started measuring extra beats at the maximum of ISO effect on CM shortening. Short-term exposure of ISO was only slightly affected by PDE4B and PDE4D ablation, whereas a long-term exposure, which could potentially better mimic pathophysiological processes associated with chronic catecholamine elevation uncovered more pronounced effects of PDE4B as compared to PDE4D. In the even more relevant Langendorff system, we observed absolutely no effect of PDE4D deletion and strong contribution of PDE4B to ISO induced arrhythmias. In both systems, the effects of PDE4B ablation on arrhythmia susceptibility could also potentially originate from both altered regulation of the LTCC complex and from direct effect on the local cAMP levels at and the leakiness of RyR2. Collectively, our data suggest that PDE4B can physically associate with and predominantly regulates cAMP levels in the RyR2 microdomain, as well as arrhythmia susceptibility upon prolonged catecholamine exposure.

## 5. Conclusions

In conclusion, our data demonstrate that the microdomains formed around Ca^2+^ handling proteins in adult mouse CMs are differentially regulated by PDE4B and PDE4D. Systematic functional live cell imaging analysis using differentially targeted FRET-based cAMP biosensors led to the new finding that while both PDE4 subfamilies participate in the cAMP regulation at the caveolin-rich plasma membrane, PDE4B is predominantly involved in the regulation of cAMP at the RyR2. This adds to the well documented previous finding that cAMP at SERCA2a is under exclusive control by PDE4D. Therefore, even within one organelle such as SR, at least two distinct PDE4 regulated cAMP microdomains can co-exist, suggesting a more complex than previously thought regulatory pattern with multiple PDEs acting at the same location.

## Figures and Tables

**Figure 1 cells-13-00476-f001:**
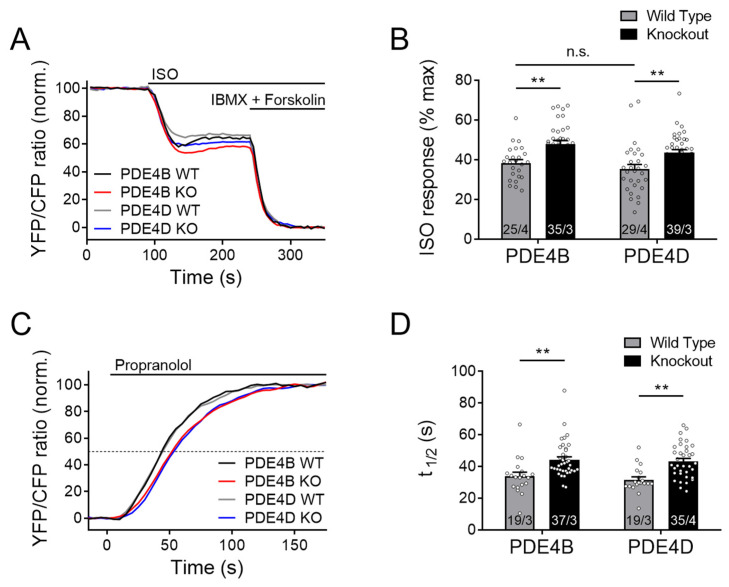
Both PDE4B and PDE4D regulate cAMP signaling events in the caveolin-rich plasma membrane upon β-adrenergic stimulation. FRET experiments with adult mouse cardiomyocytes (CMs), freshly isolated from PDE4B-WT, PDE4B-KO, PDE4D-WT and PDE4D-KO mice harboring the caveolin-rich plasma membrane targeted FRET biosensor pmEpac1. (**A**) Averaged FRET traces (25/4 for PDE4B WT, 35/3 for PDE4B KO, 29/4 for PDE4D WT and 39/3 for PDE4D KO) of CMs stimulated with 100 nmol/L Isoproterenol (ISO) followed by 100 µmol/L 3-isobutyl-1-methylxanthine (IBMX) and 10 µmol/L Forskolin. (**B**) Quantification of ISO responses as a % of maximal cAMP response. (**C**) Averaged FRET traces from n cells/N hearts (19/3 for PDE4B WT, 37/3 for PDE4B KO, 19/3 for PDE4D WT and 35/4 for PDE4D KO) of 100 nM ISO prestimulated CMs treated with 100 µmol/L propranolol. (**D**) Quantification of τ_1/2_ for propranolol induced local cAMP hydrolysis. Data from n cells/N mice are presented as means ± SEM, **—significant differences at *p* < 0.01 by mixed ANOVA followed by Wald χ^2^ test, n.s.—not significant by the same test.

**Figure 2 cells-13-00476-f002:**
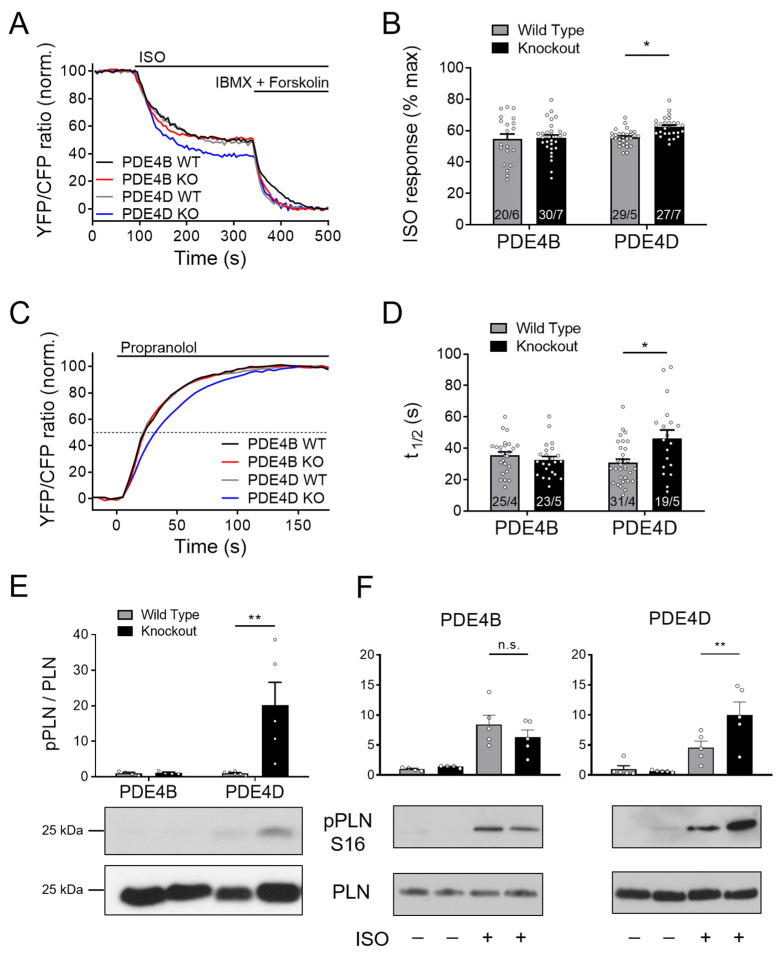
PDE4D regulates cAMP signaling in the SERCA2a microdomain. (**A**–**D**) FRET experiments with adult mouse cardiomyocytes (CMs) freshly isolated from PDE4B-WT, PDE4B-KO, PDE4D-WT and PDE4D-KO mice harboring the SERCA2a microdomain targeted biosensor Epac1-PLN. (**A**) Averaged FRET traces from n cells/N hearts (20/6 for PDE4B WT, 30/7 for PDE4B KO, 29/5 for PDE4D WT and 27/7 for PDE4D KO) of CMs stimulated with 100 nmol/L Isoproterenol (ISO) followed by 100 µmol/L 3-isobutyl-1-methylxanthine (IBMX) and 10 µmol/L Forskolin. (**B**) Quantification of ISO response as a % of maximal cAMP response. (**C**) Averaged FRET traces (25/4 for PDE4B WT, 23/5 for PDE4B KO, 31/4 for PDE4D WT and 19/5 for PDE4D KO) of 100 nmol/L ISO prestimulated CMs treated with 100 µmol/L propranolol. (**D**) Quantification of τ_1/2_ in propranolol treated CMs. FRET data from n cells/N mice are presented as means ± SEM, *—significant differences at *p* < 0.05 by mixed ANOVA followed by Wald χ^2^ test. (**E**) Representative immunoblots for PLN phosphorylation at Ser-16 in left ventricular tissues acutely isolated from PDE4B-WT, PDE4B-KO, PDE4D-WT and PDE4D-KO mice with quantification of Ser-16 phosphorylation relative to wildtype tissues and total PLN levels. (**F**). Analysis of PLN phosphorylation at Ser-16 in CMs isolated from PDE4B-KO, PDE4D-KO and respective wildtype littermates and stimulated for 10 min with 10 nmol/L ISO. Data in E and F are means ± SEM, *n* = 4–6 hearts per group. **—significant difference at *p* < 0.01 by one-way ANOVA followed by Sidak multiple comparison test. n.s.—not significant.

**Figure 3 cells-13-00476-f003:**
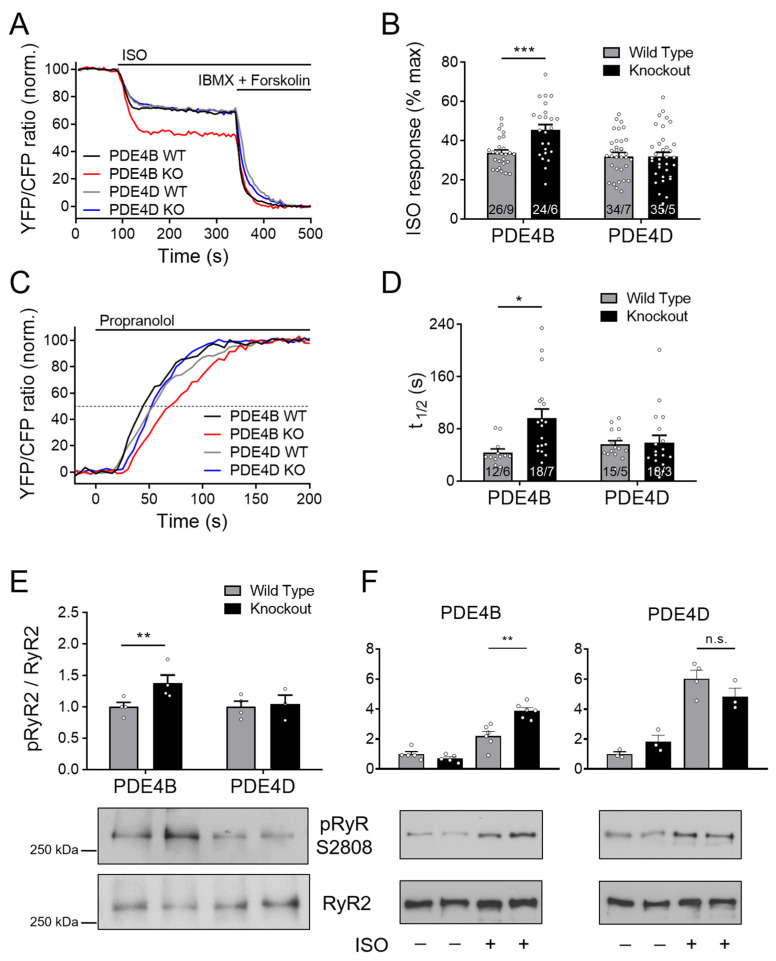
PDE4B regulates cAMP signaling in the RyR2 microdomain. FRET experiments with adult mouse cardiomyocytes (CMs) freshly isolated from PDE4B-WT, PDE4B-KO, PDE4D-WT and PDE4D-KO mice harboring the RyR2 microdomain targeted FRET biosensor Epac1-JNC. (**A**) Averaged FRET traces from n cells/N hearts (26/9 for PDE4B WT, 24/6 for PDE4B KO, 34/7 for PDE4D WT and 35/5 for PDE4D) recorded from CMs stimulated with 100 nmol/L Isoproterenol (ISO) followed by 100 µM 3-isobutyl-1-methylxanthine (IBMX) and 10 µmol/L Forskolin. (**B**) Quantification of ISO responses compared to maximal cAMP response. (**C**) Averaged FRET traces (12/6 for PDE4B WT, 18/7 for PDE4B KO, 15/5 for PDE4D WT and 18/3 for PDE4D KO) from 100 nmol/L ISO prestimulated CMs treated with 100 µmol/L propranolol. (**D**) Quantification of τ_1/2_ in propranolol treated CMs. FRET data from n/N experiments are presented as means ± SEM for n measured cells isolated from N mice. *, ***—significant differences at *p* < 0.05, *p* < 0.001 by mixed ANOVA followed by Wald χ^2^ test. (**E**) Representative immunoblots for Ser-2808 phosphorylation of RyR2 left ventricular tissues acutely isolated from PDE4B-WT, PDE4B-KO, PDE4D-WT and PDE4D-KO mice. (**F**). Analysis of RyR2 phosphorylation in CMs isolated from PDE4B-KO, PDE4D-KO and respective wildtype littermates and stimulated for 10 min with 10 nmol/L ISO. Ser-2808 phosphorylation in (**E**,**F**) was quantified relative to wildtype tissues or cells and total RyR2 levels for *n* = 3–6 hearts per group, **—significant difference at *p* < 0.01, n.s—not significant, one-way ANOVA followed by Sidak multiple comparison test.

**Figure 4 cells-13-00476-f004:**
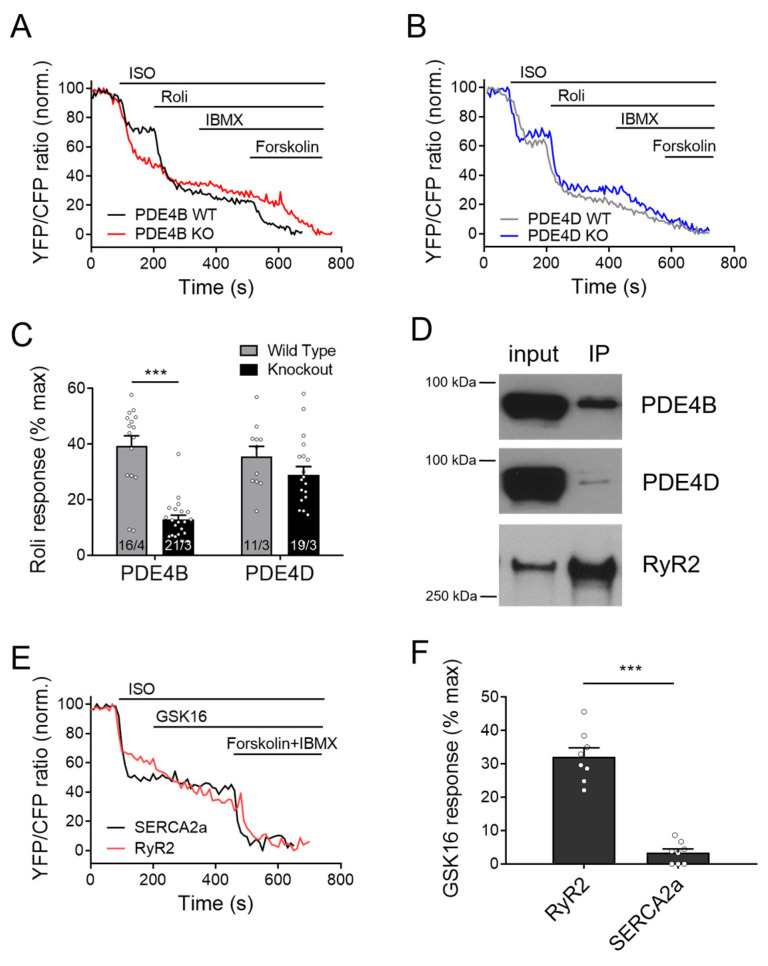
Responses to rolipram in the RyR2 microdomain of PDE4B and PDE4D-KO myocytes. FRET experiments with adult mouse cardiomyocytes (CMs) freshly isolated from PDE4B-WT and PDE4B-KO (**A**) vs. PDE4D-WT and PDE4D-KO (**B**) mice harboring the RyR2 microdomain targeted FRET biosensor Epac1-JNC and stimulated first with 100 nmol/L ISO and then by 10 µmol/L rolipram (Roli), followed by 100 µmol/L IBMX and 10 µmol/L Forskolin. Representative FRET traces (**A**,**B**) and quantification of rolipram responses as a % of maximal cAMP response (**C**). FRET data from n/N experiments are presented as means ± SEM for n measured cells isolated from N mice. ***—significant difference at *p* < 0.001 by mixed ANOVA followed by Wald χ^2^ test. (**D**) Epac1-JNC biosensor expressing heart lysates (input) were immunoprecipitated using total GFP-Trap, and IP fractions were probed for PDE4B and PDE4D. Both PDE4 subfamilies could be co-immunoprecipitated with cardiac RyR2. Representative immunoblots for *n* = 4 hearts are shown. (**E**,**F**) Measurements of the selective PDE4B inhibitor response (GSK16, 100 nmol/L) applied after 100 nmol/L ISO in Epac1-PLN (SERCA2a) or Epac1-JNC (RyR2 targeted sensor) in wildtype CMs. A total of 8 cells from 3 mice each were measured and revealed significantly stronger inhibitor response at the RyR2 as compared to SERCA2a, ***—*p* < 0.001 by mixed ANOVA followed by Wald χ^2^ test.

**Figure 5 cells-13-00476-f005:**
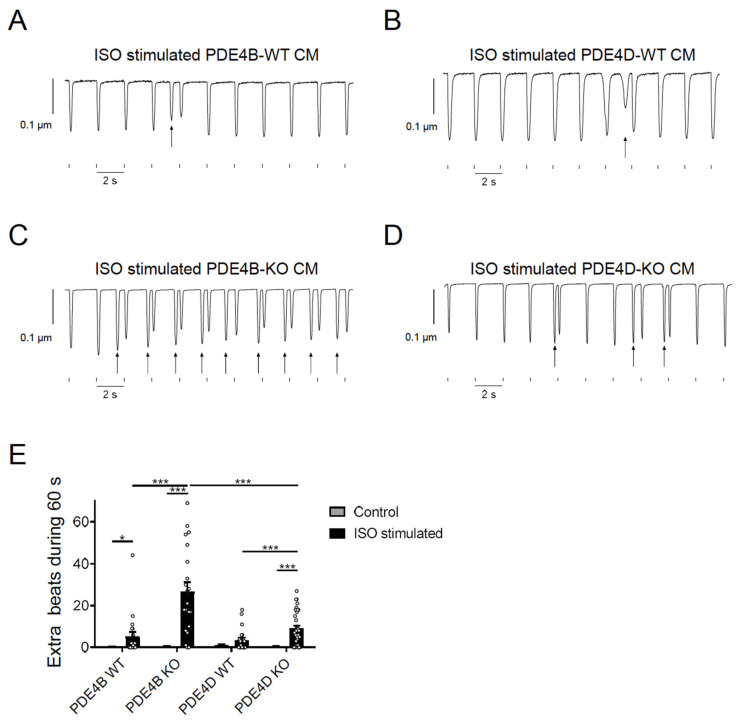
PDE4B and PDE4D regulate arrhythmia susceptibility in isolated myocytes. (**A**–**D**) Representative traces of contractility measurements in 100 nmol/L ISO treated PDE4B-WT/KO and PDE4D-WT/KO CMs. Single CMs were paced for 4 min at 0.5 s^−1^ and 15.0 V. Extra beats are marked with arrows. Traces for untreated CMs are shown in Appendix A. (**E**) Quantification of extra beats during 60 s of contractility measurement of untreated and ISO stimulated CMs. Data of contractility measurements of 20–35 individual CMs, isolated from 3–4 mice are presented as mean ± SEM. *, ***—significant differences at *p* < 0.05 and *p* < 0.001, mixed ANOVA followed by Wald χ^2^ test.

**Figure 6 cells-13-00476-f006:**
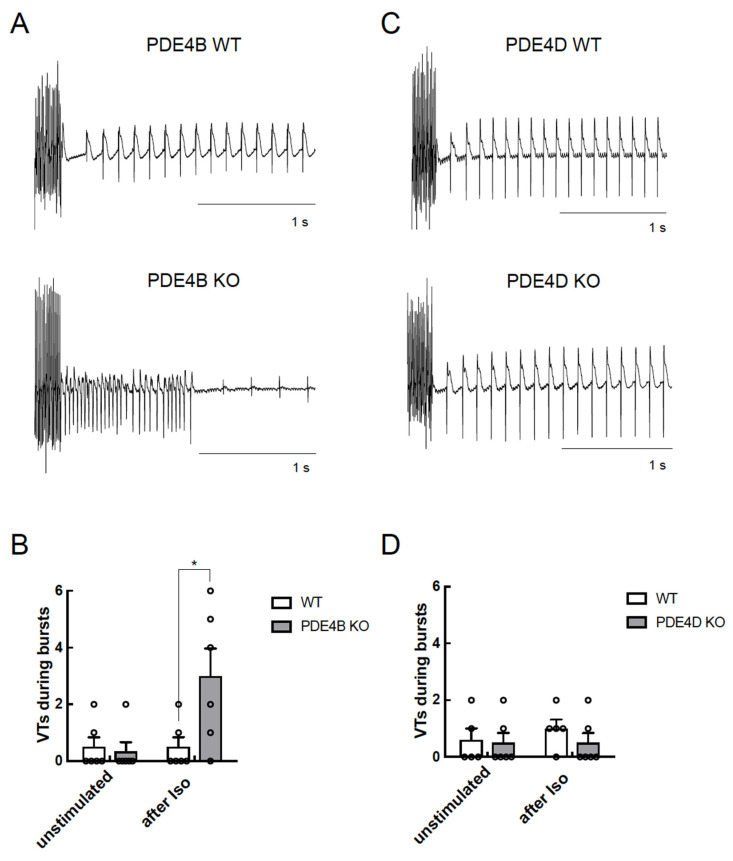
PDE4B but not PDE4D affects arrhythmia occurrence in Langendorff perfused hearts. Hearts were equilibrated for 15 min and stimulated for 10 min with 100 nmol/L ISO before a complex stimulation protocol was applied as described in Methods. Shown are representative right ventricular electrograms tracings from PDE4B WT vs. PDE4B KO (**A**,**B**) and PDE4D WT vs. PDE4D KO hearts (**C**,**D**) along with the analysis of arrhythmia occurrence which was measured as number of ventricular tachycardia (VTs) in each measured heart (*n* = 5–6 mice per group) directly after pacing protocol. Representative right ventricular electrograms for unstimulated hearts are shown in Appendix A. *—significant difference by Kruskal–Wallis test.

**Figure 7 cells-13-00476-f007:**
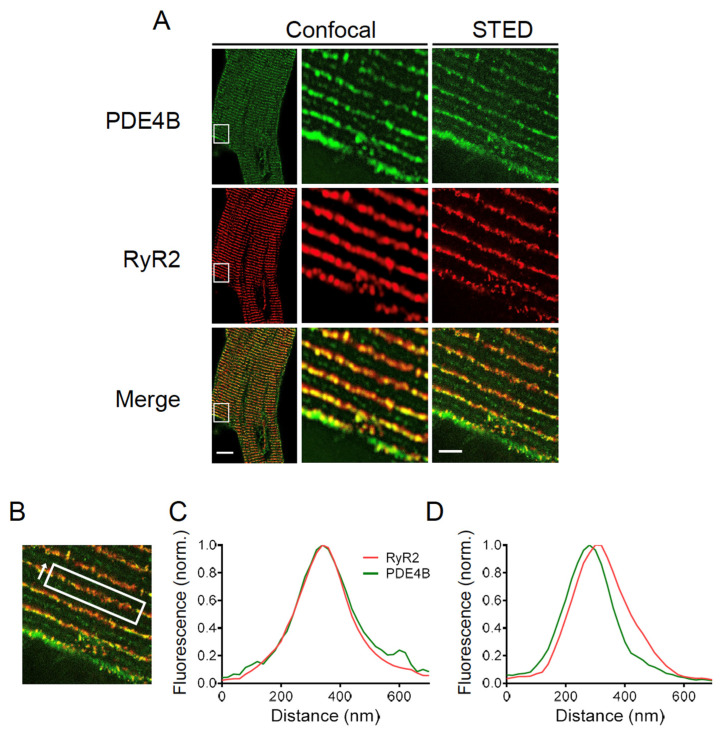
PDE4B is located in close proximity of RyR2. (**A**) Confocal and stimulated emission depletion (STED) microscopy images of a cardiomyocyte (CM) isolated of an adult wildtype mouse stained with anti-PDE4B and anti-RyR2 antibodies. Scale bars: 10 µm for whole cell image, 2 µm for STED image. (**B**) Representative region of interest for calculating the fluorescence intensity including the measurement direction. (**C**) Representative diagrams for a direct colocalization of PDE4B and RyR2. Fluorescence intensity way measured in the direction of arrow shown in (**B**). (**D**) Representative diagrams for a shifted colocalization of PDE4B and RyR2 by 20–60 nm. For (**B**–**D**), the total of 84 Z-lines from 11 individual cells isolated out of 3 different hearts were analyzed. The specificity of the PDE4B staining was confirmed in PDE4B-KO CMs, see Appendix A.

**Figure 8 cells-13-00476-f008:**
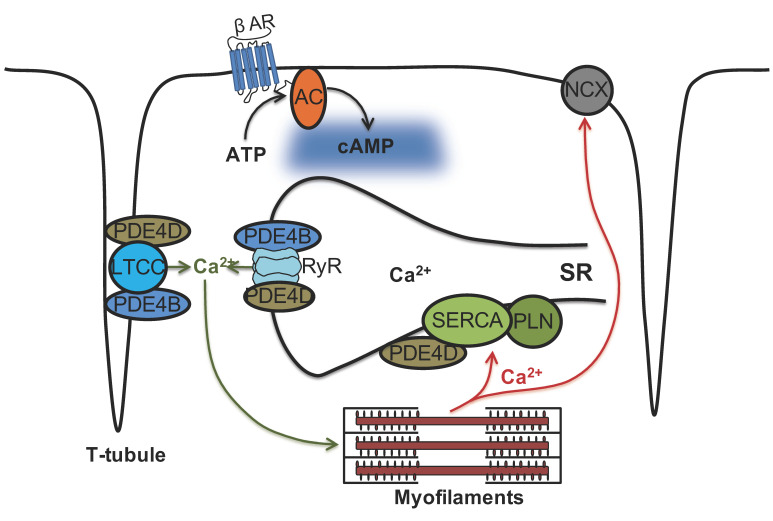
Schematic representation of phosphodiesterase subfamilies PDE4B and PDE4D and their association with cardiac calcium handling microdomains. Ca^2+^ cycling is crucial for a proper cardiac excitation-contraction coupling. Our study indicates that both PDE4B and PDE4D control local cAMP levels in the caveolin-rich plasma membrane microdomains some of which, especially T-tubules, are responsible for Ca^2+^ influx via LTCC. In contrast, SERCA2a microdomain which is responsible for diastolic Ca^2+^ reuptake is under predominant control by PDE4D. Strikingly, both PDE4B and PDE4D physically associate with and regulate local cAMP levels in the vicinity of the Ca^2+^ release channel RyR2 but PDE4B takes the predominant control over this microdomain. Hence, there are at least two differentially controlled endoplasmic/sarcoplasmic reticulum (SR) cAMP microdomains regulating calcium cycling, providing a more complex picture than previously thought.

## Data Availability

Data are contained within the article or Appendix A. Raw data and materials are available from the authors upon reasonable request.

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
