# Peer review of "Phosphodiesterases 4B and 4D Differentially Regulate cAMP Signaling in Calcium Handling Microdomains of Mouse Hearts"

_cells, 2024, doi:10.3390/cells13060476_

Round 1

Reviewer 1 Report

Comments and Suggestions for Authors

The study by Kraft at al systematically analyzed the contribution of PDE4B and PDE4D to the regulation of cAMP dynamics in three distinct subcellular microdomains. Authors elegantly exploited mice expressing FRET-based sensors for cAMP targeted to the caveolin-rich plasma membrane domain, the sarcoplasmic reticulum (SR) Ca2+-ATPase (SERCA2a) compartment and cardiac ryanodine receptor type 2 (RyR2) domain, crossed with PDE4B and PDE4D knock out animals. By performing FRET on adult cardiomyocytes, they found that both isoforms regulate cAMP in the caveolin-rich compartment, while PDE4D mainly controls cAMP dynamics around SERCA2a and, unexpectedly, PDE4B predominantly regulates cAMP in the vicinity of RyR2. Finally, authors showed that genetic inactivation of PDE4B but not PDE4D increases arrhythmias occurrence in Langendorff-perfused hearts.

Overall, the manuscript is very well presented and pleasant to read. However, this reviewer has the impression that authors’ conclusions are not always supported by the experimental data.

Major points:

1)      Interpretation of data from PDE4B and PDE4D knock out cells could be confounded by compensatory upregulation of the other isoform. Although in  Suppl. Fig. 1 authors showed no compensatory changes in the protein expression levels of the other isoform, one might not exclude that the catalytic activity is different. This should be proven experimentally.

2)      In Figure 2A-B, the effect of PDE4D knock-out on FRET signals is minor. This might be indicative of the redundancy of the 4B and 4D isoforms. Since it is not fair to ask the authors to generate double knock out cells, they might consider performing an experiment in PDE4D KO cardiomyocytes in the presence of a specific PDE4B inhibitor.

3)      Although I understand that the finding that PDE4B but not PDE4D controls RyR2-centered cAMP dynamics contrasts with previous reports (ref [20]), the present study clearly excludes the involvement of PDE4D at the RyR2 cAMP microdomain. Figure 4 clearly shows no effect of PDE4D inactivation, thus supporting the exclusive role of PDE4B. Data reported in Figure 6 further reinforces this model. Although Figure 4D shows that both PDE4B and PDE4D co-immunoprecipitated with RYR2, the amount of 4D is minor and might not be specific (this experiment lacks a negative control, that is IP from KO hearts/cells). Thus, the authors’ conclusion that “PDE4B>PDE4D regulate cAMP signaling in the RyR2 microdomain” should be modified accordingly as, from the experimental data provided, I cannot see an involvement, even minor, of PDE4D.

4)      Figure 7: This data could be strengthened by presenting STED microscopy images of PDE4B KO hearts/cells.

Minor points:

1)      Figure 2E and 3E: It is unclear if the PLN WB is relative to untreated or ISO-treated cells.

2)      Ref. 18 is inappropriate as it does not refer to “a previous report showing that both PDEs co-immunoprecipitate with LTCC [18]”.

3)      Line 502: “PDE4B an important role”. The verb is missing.

Author Response

We would like to thank the Reviewer for their very positive evaluation of our manuscript. We have done all requested additional experiments as pointed out in the point-by-point response below. The changes made to the manuscript text are underlined.

Major points:

1)      Interpretation of data from PDE4B and PDE4D knock out cells could be confounded by compensatory upregulation of the other isoform. Although in Suppl. Fig. 1 authors showed no compensatory changes in the protein expression levels of the other isoform, one might not exclude that the catalytic activity is different. This should be proven experimentally.

We thank the Reviewer for raising this important point. To address this issue experimentally, we included data from the biochemical PDE activity assay which measures catalytic activity of the PDE4 subfamily enzymes (4B and 4D) in immunoprecipitates from WT, PDE4B and PDE4D-KO hearts. We could not see major compensation by other PDE4 subfamily in individual knockouts. In PDE4D-KO mice, there was no significant change of PDE4B activity. In PDE4B-KO hearts, there was a slight but significant increase in PDE4D activity which had unlikely any impact on our findings since at the cAMP levels we could not observe higher e.g. PDE4D contributions in PDE4B-KO CMs in neither SERCA2a nor in RyR2 microdomains. We included these data into the new Supplementary Figure 1G and discussed them in the text on page 5.

2)      In Figure 2A-B, the effect of PDE4D knock-out on FRET signals is minor. This might be indicative of the redundancy of the 4B and 4D isoforms. Since it is not fair to ask the authors to generate double knock out cells, they might consider performing an experiment in PDE4D KO cardiomyocytes in the presence of a specific PDE4B inhibitor.

We thank the Reviewer for this nice suggestion. As requested, we have now preformed new measurements using the selective PDE4B inhibitor GSK16. Please see the new Figure 2F (and also the new Figures 4E/F). In figure 2F, we followed checked for the redundancy between PDE4B and PDE4D by treating PDE4D-KO cells with the PDE4B selective inhibitor. We could not observe significant effect of the PDE4B inhibitor on the amplitude of ISO response from the SERCA2a targeted sensor, suggesting a low level of redundancy. In Figure 4E/F, we performed another experiments suggested by the Reviewer 3 where we compared the effect of GSK16 applied to wildtype myocytes expressing either SERCA2a or RyR2 targeted cAMP biosensor and found that the inhibitor response was much stronger at the RyR2 receptor as compared to SERCA2a. Please, see this new figure and the respective new text on page 10. 

3)      Although I understand that the finding that PDE4B but not PDE4D controls RyR2-centered cAMP dynamics contrasts with previous reports (ref [20]), the present study clearly excludes the involvement of PDE4D at the RyR2 cAMP microdomain. Figure 4 clearly shows no effect of PDE4D inactivation, thus supporting the exclusive role of PDE4B. Data reported in Figure 6 further reinforces this model. Although Figure 4D shows that both PDE4B and PDE4D co-immunoprecipitated with RYR2, the amount of 4D is minor and might not be specific (this experiment lacks a negative control, that is IP from KO hearts/cells). Thus, the authors’ conclusion that “PDE4B>PDE4D regulate cAMP signaling in the RyR2 microdomain” should be modified accordingly as, from the experimental data provided, I cannot see an involvement, even minor, of PDE4D.

We fully agree with the Reviewer that PDE4B plays a major if not exclusive role in the control of cAMP at the RyR2. In the first manuscript we tried to be formally polite and not to rule out the role of PDE4D completely. We have now rephrased the text and modified our conclusion in full accordance with the Reviewer’s suggestion. Re the negative CoIP control, we could provide one for PDE4B in the new Supplementary figure 3B, whereas the one for PDE4D was inconclusive (therefore, we decided not to include it into the manuscript).

4)      Figure 7: This data could be strengthened by presenting STED microscopy images of PDE4B KO hearts/cells.

Thank you for this important suggestion. We have performed the same experiments with PDE4B-KO CMs and could confirm the PDE4B staining to be specific. We include these negative KO control images into the new Supplementary Figure 6. Since the PDE4B signal was very weak (almost completely abolished as expected) which was not sufficient for additional STED imaging, we have shown only regular confocal images.

Minor points:

1)      Figure 2E and 3E: It is unclear if the PLN WB is relative to untreated or ISO-treated cells.

Thank you, we added this info to the figure legends.

2)      Ref. 18 is inappropriate as it does not refer to “a previous report showing that both PDEs co-immunoprecipitate with LTCC [18]”.

Sorry for this mistake. It is shown in the Ref. 19, so we have fixed it accordingly.

3)      Line 502: “PDE4B an important role”. The verb is missing.

Fixed as suggested.

Reviewer 2 Report

Comments and Suggestions for Authors

In this manuscript, Kraft et al. conducted an interesting study on the role of two different subtypes of phosphodiesterase 4 (PDE4B and PDE4D) in murine cardiomyocytes to investigate their implications in Ca2+ signaling. Considering the importance of cAMP compartmentalization, three different Ca2+ handling microdomains were studied.

Through an elegant combination of FRET, single-cell contractility assessments, immunofluorescence, immunoblotting, co-immunoprecipitation, and Langendorff-perfused hearts, these authors managed to discover that while both PDE4 subfamilies participate in cAMP regulation at the caveolin-rich plasma membrane, PDE4B is more important at the RyR2 receptors, and SERCA2a is exclusively regulated by PDE4D.

The manuscript is well-written, the methods are well-described, and the discussion is well-structured according to the results. Graphs and statistical treatment of data are clear, and the conclusions make this paper an interesting source of information for researchers working on calcium signaling and cardiac arrhythmias.

Author Response

We would like to thank the Reviewer for their positive evaluation of our manuscript and for a highly favourable assessment. 

Reviewer 3 Report

Comments and Suggestions for Authors

In the manuscript “Phosphodiesterases 4B and 4D differentially regulate cAMP signaling in calcium handling microdomains of mouse hearts” the authors, by using transgenic mice expressing FRET-based cAMP-specific biosensors targeted to caveolin reach, SERCA2A and RYR2 microdomains crossed to PDE4B and PDE4D knockout mice, investigate the contribution of these PDEs in the confinement of cAMP microdomains. Briefly, the authors assert that PDE4B controls mostly cAMP level around RyR2 and PDE4D  regulates SERCA2a cAMP microdomain.

Few point should be clarified to improve the manuscript

1.     PDE4BWT and PDE4DWT seem to have a different ISO response. Was a statistical analysis carried out between PDE4BWT and PDE4DWT groups? (Figure 1A-B) .  

2.    Please include immunoblots for PLN phosphorylation at Ser-16 in myocytes isolated from PDE4B-WT, PDE4B-KO, PDE4D-WT and PDE4D-KO after ISO stimulation given that the differences in FRET experiments are appreciable after b-adrenergic stimulation (Figure 2E) and the same for pRYRS2808 (Figure 3E). Moreover I see just 3 dots in the dot plot for PDE4DWT and 2 for PDE4DKO, can you explain this issue? Finally In the manuscript https://doi.org/10.1161/CIRCRESAHA.111.250464Circulation Research. 2011;109:1024–1030 Data regarding increased pPLB phosphorylation in PDE4DKO have been already published, same for pRyR2808. In the latter case it is diminished in PDE4DKO mouse. First, you should cite the manuscript. Second, you should explain these discrepancies. 

3. Would be helpful to perform some experiments using specific PDE4B inhibitors.

Author Response

We would like to thank the Reviewer for their positive evaluation of our manuscript. We have done all requested additional experiments as pointed out in the point-by-point response below. The changes made to the manuscript text are underlined.

Few point should be clarified to improve the manuscript

  1. PDE4BWT and PDE4DWT seem to have a different ISO response. Was a statistical analysis carried out between PDE4BWT and PDE4DWT groups? (Figure 1A-B).

We thank the Reviewer for raiding this point. We have now included the comparison between the PDE4BWT and PDE4DWT groups in our statistical analysis and found no significant different. This has now been annotated in the revised Figure 1B.

  1. Please include immunoblots for PLN phosphorylation at Ser-16 in myocytes isolated from PDE4B-WT, PDE4B-KO, PDE4D-WT and PDE4D-KO after ISO stimulation given that the differences in FRET experiments are appreciable after b-adrenergic stimulation (Figure 2E) and the same for pRYRS2808 (Figure 3E). Moreover I see just 3 dots in the dot plot for PDE4DWT and 2 for PDE4DKO, can you explain this issue? Finally, In the manuscript https://doi.org/10.1161/CIRCRESAHA.111.250464Circulation Research. 2011;109:1024–1030 Data regarding increased pPLB phosphorylation in PDE4DKO have been already published, same for pRyR2808. In the latter case it is diminished in PDE4DKO mouse. First, you should cite the manuscript. Second, you should explain these discrepancies.

We would like to thank the Reviewer very much for suggesting to show immunoblot data for ISO-stimulated myocytes. Indeed, we have shown only basal phosphorylation data measured in freshly isolated myocytes lysed right away after cell isolation. This makes the data more comparable to some published literature, e.g. Beca et al 2011 paper cited by the Reviewer. We have also done some immunoblots with ISO stimulated Langendorff perfused hearts which showed essentially the same result. Since these experiments were done using a different protocol in an earlier phase of the study, we decided not to include them into the manuscript:

please see Figure for Reviewer in the pdf version of this responce attached...

We apologise for the dots in Figure 3E which appeared small and partially overlapped with the black coloured bars. We have now increased the size of the dots and applied white background to them which helps increase their visibility.

Finally, we have commented on the immunoblots data published in Beca et al. 2011 (our Ref. 37) and compared our results with this publication (adding to the discussion section on page 14), which was very helpful to emphasize how compatible our data actually are with already published ones. Thank you.

  1. Would be helpful to perform some experiments using specific PDE4B inhibitors.

As suggested, also by Reviewer 1, we have included some measurements with the selective PDE4B inhibitor GSK16 into the figure 3 and 4 of the manuscript. Please see the new Figures 2F and 4E/F. In figure 2F, we followed Reviewer 1 suggestion to check for the redundancy between PDE4B and PDE4D by treating PDE4D-KO cells with the PDE4B selective inhibitor. We could not observe significant effect of the PDE4B inhibitor (established GSK16 compound) on the amplitude of ISO response from the SERCA2a targeted sensor. In the new Figure 4E/F, we have compared the effect of GSK16 applied to wildtype myocytes expressing either SERCA2a or RyR2 targeted cAMP biosensor and found that the inhibitor response was much stronger at the RyR2 receptor as compared to SERCA2a. Please, see this new figure and the respective new text on page 10.  

Round 2

Reviewer 1 Report

Comments and Suggestions for Authors

Authors have satisfactorily addressed all the concerns raised by this Reviewer.

Author Response

We thank the Reviewer for the positive evaluation of our revised manuscript.

Reviewer 3 Report

Comments and Suggestions for Authors

Author Response

  1. Actually, the results included in authors response are not comparable to what presented in the manuscript since the basal level of pPLB in PDE4DKO mice are not increased compared to PDE4DWT and the wb provided are normalized for GAPDH and not for PLB. Authors should make this effort and provide WB of freshly isolated myocytes treated with ISO.

We thank the Reviewer for raising this point. As suggested we performed immunoblots with isolated myocytes treated with ISO and included them into the new figure 2F and new figure 3F. The results we obtained were very well compatible with FRET results - we see increase RyR phosphorylation in 4B-KO and increased PLN phosphorylation in 4D-KO cells. In contrast to acutely isolated heart tissues in figures 2E/3F (which are subjected to in vivo catecholamine effects), there is no increase in basal phophorylation in isolated cells which was expected. We must apologise that only during the revision we noticed that there was a mistake in the description of figures 2E/2F which show experiments looking at basal PLN/RyR phosphorylation performed with acutely isolated and snap frozen left ventricular tissues and not in cells. This came from the older version of the manuscript and was unfortunately overlooked at previousl ocasions. 

2. Regarding figure 4E/F is not clear to me why GSK16 was applied after ISO treatment, moreover in figure 4E trace differences after GSK16 treatment between Serca2a and  RYR2 targeted biosensors do not correspond to what shown in figure 4F. More representative traces should be included.

Thank you for this question. In figure 4, we showed GSK16 responses after ISO which are better comparable with Rolipram responses shown in the same figure 4A-C. Regarding the representative traces, GSK responses in these traces had an amplitude (calculated as a % max response) nicely fitting the average data on the bar graph shown in 4F. We have now better explained how these responses were calculated in the figure legend. Moreover, for better comparison with a previous figures, we also included a new supplementary figure 2D showing the results of ISO responses with and without GSK.